# Development and Validation of the Meiji Nutritional Profiling System (Meiji NPS) to Address Dietary Needs of Adults and Older Adults in Japan

**DOI:** 10.3390/nu16070936

**Published:** 2024-03-24

**Authors:** Ryota Wakayama, Adam Drewnowski, Tomohito Horimoto, Yoshie Saito, Tao Yu, Takao Suzuki, Satoshi Takasugi

**Affiliations:** 1Meiji Co., Ltd., 2-2-1 Kyobashi, Chuo-ku, Tokyo 104-9306, Japan; tomohito.horimoto@meiji.com (T.H.); tao.yu@meiji.com (T.Y.); satoshi.takasugi@meiji.com (S.T.); 2Center for Public Health Nutrition, University of Washington, Seattle, WA 98195, USA; adamdrew@uw.edu; 3Institute for Gerontology, J. F. Oberlin University, 3758 Tokiwa, Machida, Tokyo 194-0294, Japan

**Keywords:** nutrient profiling, nutrient-rich foods index, malnutrition, older adults, frailty, convergent validity, Japanese diet, hybrid nutrient density score, nutrients, food groups

## Abstract

This study introduces the Meiji Nutritional Profiling System (Meiji NPS), which was specifically designed to respond to age-related shifts in nutrient requirements among Japanese adults (<65 years old) and older adults (≥65 years old). Japan has one of the most aged societies in the world. The health issues of interest are malnutrition and lifestyle-related diseases among adults and frailty among older adults. Two versions of the NPS were developed based on nutrients to encourage (protein, dietary fibers, calcium, iron, and vitamin D), food groups to encourage (fruits, vegetables, nuts, legumes, and dairy), and nutrients to limit (energy, saturated fatty acids, sugars, and salt equivalents). The Meiji NPS for older adults did not include iron or saturated fatty acids. The algorithms were based on the Nutrient-Rich Foods Index (NRF). The convergent validity between the Meiji NPS and the existing NPSs for the same foods was confirmed using Spearman’s correlation coefficients (NRF: r = 0.67 for adults and r = 0.60 for older adults; Health Star Rating: r = 0.64 for adults and r = 0.61 for older adults). The Meiji NPS may be useful for nutritional evaluation and reformulation of food products, tailored to adults and older adults to ameliorate health issues in Japan.

## 1. Introduction

Assessing the nutrient density of foods is a challenge, given that every food product contains multiple nutrients. Foods can include protein, fiber, vitamins, and minerals, as well as fats, sugar, and sodium [1]. Nutrient profiling (NP) methods are quantitative metrics designed to capture the overall nutrient contents of foods in relation to population nutrient needs [2,3,4]. In general, those foods that best satisfy population nutrient requirements receive the highest scores [4]. However, nutrient requirements, the foundation of NP methods, may differ depending on sex and age [5,6].

The Meiji Nutritional Profiling System (Meiji NPS) is one of the first attempts to tailor NP methods to the nutrient needs of adults (<65 years old) and older adults (≥65 years old). The World Health Organization (WHO) has specifically stated that NP models need to classify or rank foods according to their nutritional composition for reasons related to disease prevention and health promotion [7]. The two health conditions specifically addressed by the Meiji NPS are different forms of malnutrition among adults and excessive frailty among older adults in Japan.

The Meiji NPS has the potential to promote healthier dietary choices and drive the reformulation of food products to better align with public health objectives. There are precedents for both approaches. NP models developed by governments are intended to encourage consumers to select healthier foods [8,9,10]. For example, the Australian Government developed the Health Star Rating (HSR) as the basis for front-of package labels. The HSR was mainly designed to address obesity as Australia has one of the highest rates of obesity in the world [11,12,13]. The HSR nutrients to encourage are protein and fiber, whereas energy, total sugar, saturated fatty acids (SFAs), and sodium are the components to limit. Among the food groups to encourage in the HSR are fruits, vegetables, nuts, and legumes. At the same time, food manufacturers have used their own NP models to reformulate their products [14,15,16,17], most notably to limit their energy, fat, sugar, and sodium contents. The two approaches are linked. The Global Access to Nutrition Index uses the HSR to motivate food manufacturers to improve the nutritional value of their products [18].

According to the WHO, NP models need to address an identified public health problem [19]. Health and nutritional issues vary based on regions, lifestyles, and life stages [20]. Among specific health issues identified for Japan are malnutrition, which manifests as underweight or excessive thinness among young women, as well as overweight [21,22,23,24,25,26,27,28,29,30,31,32]. Frailty among older adults is a particularly serious issue in Japan, which has one of the largest aging populations in the world [6,33,34]. The Meiji NPS was designed to address health issues at different life stages, taking a life-course approach to NP modeling throughout the human lifespan [20,35,36,37,38,39]. Given the super-aging society in Japan, an NP model designed for older adults was thought to be particularly useful.

The overall quality of Japanese dietary habits can be assessed in several ways [40,41,42,43]. The dietary nutrient density of older adults is a matter of special concern. As people age, they consume fewer calories daily, but their nutrient requirements remain largely the same or even increase. As a result, the diets of older adults need to incorporate more nutrients per calorie.

Existing NP models, largely intended for front-of-package labels, do not take changing nutrient needs across the life cycle into account. Health issues that vary according to life stage need to be considered, particularly those identified among older adults ≥65 years old who follow traditional eating habits in Japan. The Meiji NPS was tailored to the dietary habits, specific health issues at each life stage, and special demographic composition of the Japanese population. The objective of this study was to develop a new age-sensitive NPS for Japan and investigate its validity. We developed two versions of the Meiji NPS, one for adults and one for older adults (≥65 years old), with a focus on preventing lifestyle-related diseases, including extreme frailty in old age. The Meiji NPS may be useful for the nutritional evaluation and reformulation of food products tailored to adults and older adults to ameliorate the health issues in Japan.

## 2. Materials and Methods

### 2.1. Scope and Principles of the Meiji NPS

The Meiji NPS, developed to address age-related health issues in Japan, was used primarily to guide improvements in diet quality and the reformulation of food products. The population groups of interest were adults <65 years old and older adults ≥65 years old. Health issues were lifestyle-related diseases, such as overweight/obesity, hypertension, dyslipidemia, and type 2 diabetes in adults; thinness in young women; and frailty in older adults. The Meiji NPS was developed as follows: (1) selection of nutrients to encourage; (2) selection of food groups to encourage; (3) selection of nutrients to limit; (4) selection of Reference Daily Values (RDVs); (5) development of the Meiji NPS algorithm; and (6) validation and testing.

### 2.2. Overview of Nutrients to Encourage/Limit and Food Groups to Encourage

The Meiji NPS for adults was based on protein, dietary fiber, calcium, iron, and vitamin D as nutrients to encourage and energy, SFAs, sugar, and salt equivalents as the nutrients to limit in foods. The Meiji NPS for older adults was based on encouraging protein, dietary fiber, calcium, and vitamin D and limiting energy, SFAs, sugar, and salt equivalents in foods. Food groups to encourage were the same for both Meiji NPS versions, fruits, vegetables, nuts, legumes, and dairy.

### 2.3. Selection of Nutrients to Encourage

Based on the Dietary Reference Intakes for Japanese (2020) [6], protein, dietary fiber, calcium, iron, and vitamin D are consumed in insufficient amounts in Japan, particularly by older adults [44]. Higher protein and dietary fiber intakes are reportedly associated with a lower obesity risk [45,46,47]. Calcium intake is reportedly related to a reduction in the rates of hypertension and type 2 diabetes among adults and a reduction in the rate of dementia among older adults [48,49,50]. Vitamin D intake is reportedly associated with a lower risk of dyslipidemia and type 2 diabetes and prediabetes [51,52,53]. A low level of vitamin D is reportedly associated with age-related health issues, such as sarcopenia, frailty, cognitive functional decline, and dementia [54,55,56,57,58,59,60,61,62,63,64,65]. Iron-deficiency anemia is a serious health issue due to the desire for thinness among young women in Japan [66]. Underweight young women in Japan also exhibit a low intake of dietary fiber, calcium, iron, and vitamin D [67]. Thus, protein, dietary fiber, calcium, iron, and vitamin D were deemed the main nutrients to encourage in the Meiji NPS. Iron was excluded from the Meiji NPS for older adults owing to its adequate intake among Japanese older adults and the lesser role it plays in frailty.

### 2.4. Selection of Food Groups to Encourage

Hybrid NP scores combine both nutrients and selected food groups. The Meiji NPS was designed to include food groups that are rich in additional nutrients of interest, such as potassium and vitamin C. In alignment with this philosophy, the Meiji NPS did not establish any food groups to discourage. The selection of food groups to encourage, as guided by the “Health Japan 21” [68] and “Healthy diet” [20] guidelines, was as follows: fruits, vegetables, nuts, legumes, and dairy. In the Meiji NPS, vegetables also included mushrooms, algae, and spices. We followed the lead of a previous study, in which the foods of the Japanese Food Standard Composition Table 2020 Edition (8th Edition) were grouped in terms of nutrient density [69] by including green teas, black teas, and coffee as vegetables and cocoa (pure powder) as legumes. These foods reportedly provide benefits in terms of obesity [70,71,72,73,74,75,76,77], hypertension [78,79,80,81], dyslipidemia [78,82,83,84,85,86,87,88,89,90], and diabetes [76,77,91]. The composition of food groups to encourage followed the coding scheme of the Japanese Food Standard Composition Table 2020 Edition (8th Edition) [92]. Fruits in the Meiji NPS corresponded to fruits in that table; vegetables corresponded to vegetables, mushrooms, and algae in that table; nuts corresponded to nuts in that table; legumes corresponded to pulses in that table; and dairy corresponded to milk and milk products in that table.

### 2.5. Selection of Nutrients to Limit

The selection of nutrients to limit closely followed other NP models. Nutrients to limit were SFAs, sugars, and salt equivalents (sodium content multiplied by 2.54). Energy was included in the list of nutrients to limit. The Dietary Reference Intakes for Japanese (2020) [6] suggested that energy, SFAs, and sodium are associated with hypertension and that energy, SFAs, and sugars are risk factors for obesity. Furthermore, a reduction in sugar has an effect on body weight [93]. In addition to a strong desire to become thin and to lose weight [94], unhealthy dietary intakes have been observed among young Japanese women [95]. The consumption of some confectionaries is higher compared to normal-weight women, and the consumption of soft drinks has increased among underweight women [95,96]. Thus, these nutrients to limit for young women should be considered. The Meiji NPS defines glucose, galactose, fructose, maltose, sucrose, and lactose as sugars.

The intake of dairy products is inversely associated with frailty in older adults, and it has been suggested that moderate intake of SFAs is effective at preventing frailty in Japan [97,98]. Additionally, SFA intake is reportedly not associated with sarcopenia [99]. Thus, SFAs were excluded from the nutrients to limit in the Meiji NPS for older adults. Table 1 summarizes the principal nutrients in the Meiji NPS for adults and older adults.

### 2.6. Age-Appropriate RDVs

The two target population groups were adults (<65 years old) and older adults (≥65 years old). The age-appropriate nutrient standards are summarized in Table 2. The RDVs of protein, dietary fiber, calcium, iron, vitamin D, energy, and salt equivalents were selected according to the maximum values in the Dietary Reference Intakes for Japanese (2020) [6] for the age groups. The RDVs of SFAs and sugar were based on the WHO recommendation [100,101], the latter of which corresponds to 10% of the energy intake. The RDVs of fruits, vegetables, legumes, and dairy were based on values in “Health Japan 21” [68]. The RDV of nuts was obtained from the “EAT-Lancet planetary health diet” [102] and the “healthy diet for Japanese longevity” [103]. In the EAT-Lancet planetary health diet, the recommended median intake of peanuts is 50 g. Additionally, the tree nuts category has a separate allocation of 25 g. Therefore, the RDV of nuts was set at 75 g.

### 2.7. The Meiji NPS Algorithm

The Meiji NPS algorithm was based on nutrients and food groups to encourage, balanced against nutrients to limit (including energy). The base of calculation was 100 g, in line with the HSR. The algorithm was based on Nutrient-Rich Foods Index 9.3 (NRF9.3) [2,3,104,105,106]. The NRF9.3 was the sum of the percentage of RDVs for the nine nutrients to encourage minus the sum of percentage of RDVs for the nutrients to limit. In the original NRF9.3, calculations were made per 100 kcal (rather than 100 g) and capped at 100% (Equation (1)). The NRF is a renowned composite measure of nutrient density, and its validity has been extensively verified [104,107,108]. In addition, it is reportedly positively correlated with overall dietary quality in the Japanese population [109].
(1)$$NRF9.3=\sum_{i=1-9}\left({nutrients\;to\;encourage}_{i}/{RDV}_{i}\right)\times 100\;\;\;\;\;\;\;\;\;\;\;\;\;\;\;\;\;\;\;\;\;\;\;\;\;\;\;\;\;\;\;\;-\sum_{i=1-3}\left({nutrients\;to\;limit}_{i}/{RDV}_{i}\right)\times 100$$

The Meiji NPS score is calculated as the sum of the percentages of RDVs for the nutrients to encourage plus the sum of the percentages of RDVs for food groups to encourage minus the sum of the percentages of RDVs for the nutrients to limit (Equations (2) and (3)). The Meiji NPS score for adults was calculated per 100 g. The caps of the Meiji NPS scores for adults and older adults are summarized in Table 3. In consideration of excess intake, caps are set for nutrients and food groups to encourage but not for the nutrients to limit. Therefore, nutrients to limit can score over 100%. The percentages of protein and dietary fiber in the Meiji NPS for adults and older adults were capped at 100%. An insufficient intake is defined as the gap between the RDV and the median intake of the Japanese population [6,110]. The percentage of micronutrients (calcium, iron, and vitamin D) in the Meiji NPS for adults was capped at the gap. However, vitamin D is an important nutrient for older adults, and its intake is associated with a reduction in the risk of sarcopenia and frailty [54,55,56,57,58,59,60,61]. Therefore, the percentage of vitamin D in the Meiji NPS for older adults was capped at 100%. The percentage of foods to encourage was capped at the gaps in the Meiji NPS for adults and older adults.
(2)$$\begin{array}{ll} Meiji\;NPS\;for & adults\\ & =\sum_{i=1-5}\left({nutrients\;to\;encourage}_{i}/{RDV}_{i}\right)\times 100\\ & -\sum_{i=1-4}\left({nutrients\;to\;limit}_{i}/{RDV}_{i}\right)\times 100\\ & +\sum_{i=1-5}\left({food\;groups\;to\;encourage}_{i}/{RDV}_{i}\right)\times 100 \end{array}$$
(3)$$\begin{array}{ll} Meiji\;NPS\;for & older\;adults\\ & =\sum_{i=1-4}\left({nutrients\;to\;encourage}_{i}/{RDV}_{i}\right)\times 100\\ & -\sum_{i=1-3}\left({nutrients\;to\;limit}_{i}/{RDV}_{i}\right)\times 100\\ & +\sum_{i=1-5}\left({food\;groups\;to\;encourage}_{i}/{RDV}_{i}\right)\times 100 \end{array}$$

### 2.8. Nutrient Composition Database

Nutrient composition data came from the Japanese Food Standard Composition Table 2020 Edition (8th Edition) [92], released by the Ministry of Education, Culture, Sports, Science, and Technology, Japan. This open database lists 2478 foods and multiple macro- and micronutrients, all expressed per 100 g. As the Meiji NPS was designed for individual foods rather than meals, prepared foods were excluded, yielding a total sample size of 2428. Out of the 2428 food items, several of these had missing data, particularly for total sugar content. Total sugar content data were missing for the important category of fish and seafood (mollusks and crustaceans), as well as for meat that was either raw, minimally processed, or dried. The total sugar content for those categories was inferred in previous studies [40]. The present assumption was that the sugar content for those foods was zero [40,111,112]. For other processed foods in the fish and seafood and meat groups (e.g., fish cakes, fish boiled in soy sauce, and chicken nuggets), the amount of carbohydrates was calculated to represent the sugar content. After the inputting of missing sugar values for raw and processed seafood (n = 684 foods), a total of 1545 foods with complete data were available for analysis.

Foods were assigned into desirable food categories based on the product name in the food composition tables. Based on the food names, the foods described as cereals, potatoes and starches; sugars and sweeteners; fish and seafood; meat; eggs; fats and oils; and confectioneries were not the food groups to encourage. In contrast, items in the fruit category (except for juice-based beverages and nectars) were assumed to be 100% fruit. Juice-based beverages and nectar were assigned a 30% fruit value. The vegetable category in the Meiji NPS included vegetables, mushrooms, algae, and spices. Items in these categories were assumed to be 100% vegetables, unless it was clear from the product name that the product did not have 100% vegetable or fruit content. In general, nuts were deemed to be 100% nuts, pulses were deemed to be 100% pulses, and dairy products were deemed to be 100% dairy, with the exception of cream.

### 2.9. Statistical Analysis

Medians (interquartile ranges [IQRs]) and means ± standard deviations were used to express data. The convergent validity of the Meiji NPS for adults and older adults was tested with reference to NRF9.3. The RDVs for NRF9.3 were calculated according to the Dietary Reference Intakes for Japanese (2020) [6]. Spearman’s correlation [113] was used for comparisons between the Meiji NPS and NRF9.3 or HSR. All analyses in this study were performed using R software version 4.3.1 (The R Foundation for Statistical Computing, Vienna, Austria).

## 3. Results

### 3.1. The Meiji NPS for Adults and Older Adults

Complete data regarding nutrients and food groups were available for 1545 foods. Meiji NPS scores were calculated for adults and older adults and compared to NRF9.3 scores for the same items. Histograms displaying score distributions are presented in Figure 1 and Figure 2. The Meiji NPS score for adults ranged from −760 to 292, and that for older adults ranged from −770 to 268. These results are summarized in Table 4 and Table 5. The mean Meiji NPS score for adults and older adults was 38.9 and 39.3, respectively. The median Meiji NPS score for adults and older adults was 36.7 and 31.2, respectively. No foods were classified as sugars and sweeteners owing to the lack of nutrient data; thus, the mean, standard deviation, and median values for this item were not available.

### 3.2. Convergent Validity between the Meiji NPS and NRF9.3

The convergent validity was assessed using the Meiji NPS and NRF9.3. Spearman’s correlation coefficients between the Meiji NPS for adults or older adults and NRF9.3 were 0.67 and 0.60, respectively (Table 6). In the Meiji NPS for adults and older adults, the correlation coefficients of mushrooms, algae, and fish and seafood were relatively low and not significant (mushrooms: r = 0.14 for adults and r = 0.07 for older adults; algae: r = 0.24 for adults and r = 0.20 for older adults; fish and seafood: r = 0.07 for adults and r = −0.05 for older adults; all *p* > 0.05).

### 3.3. Convergent Validity of the Meiji NPS with HSR

Meiji NPS scores were compared to HSR scores for the same foods to provide additional validation of the convergent validity of the Meiji NPS. The HSR score was calculated using the Health Star Rating Calculator [114]. The HSR evaluates foods along a 10-point scale. The Meiji NPS scores were accordingly split into deciles. Spearman’s correlation [113] was used to compare these systems.

Spearman’s correlation coefficients were 0.64 for adults and 0.61 for older adults (Table 7). All foods categorized as algae by the HSR scored 10 points (5 stars). Scores of all foods categorized as fats and oils by the Meiji NPS for adults were 10 percentiles. Thus, the correlation coefficients were not available for these two items. The correlation coefficients for eggs were relatively low and not significant (r = −0.36 for adults and r = −0.37 for older adults; all *p* > 0.05).

## 4. Discussion

This is the first NPS that is age-dependent and focused on nutrient requirements of older adults ≥65 years old. The goal of this study was to address changing nutrient requirements at different stages of the life cycle. Some of the Meiji NPS scores differed between adults and older adults. The mean Meiji NPS score for meat was 5.3 for adults and 17.3 for older adults. The corresponding IQR shifted from −15.3–33.7 to 5.7–30.3. In a previous study, meat intake was negatively associated with frailty [98]. In addition, the Meiji NPS score for cheese differed greatly between adults and older adults. The mean score for cheese for adults was 56.2, whereas that for older adults was 80.1. The IQR shifted from 41.1–71.1 to 50.7–98.9. Previous studies revealed that the consumption of cheese presents health benefits in older adults [115,116,117,118,119]. These results indicate that the Meiji NPS for older adults is more suitable for the assessment of foods that prevent frailty than the Meiji NPS for adults.

The Meiji NPS was specifically designed to address identified health issues, and we plan to validate the relationships between diets consistent with the Meiji NPS and the selected health outcomes. The Meiji NPS was developed deductively by referring to the existing NPS, Health Japan 21, the Dietary Reference Intakes for Japanese (2020), and epidemiological studies. We set the RDVs according to the actual nutrient intake of the Japanese population and the national standards, in response to age-dependent health issues in Japan. Moreover, we selected nutritional factors and food factors related to health issues based on the actual intake of the Japanese population. The Meiji NPS is focused on the health issues of Japanese adults and older adults. However, the same approach can be applied to different countries or different age groups. When the health targets were changed, the algorithm of the Meiji NPS could be applied for other life stages, for example, children <12 years old. 

In this study, the nutritional values in the Meiji NPS were evaluated per 100 g. For adults, the foods scoring under −200 were seasonings and butter. Japanese people do not usually consume these items in quantities of 100 g at a time. Indeed, it is necessary to develop a scoring system that reflects the actual dietary habits. In terms of epidemiological studies, both food quality and quantity may affect health outcomes [120,121,122,123,124,125]. For example, a higher intake of comfort foods may be associated with rates of obesity or overweight [126,127]. However, a moderate intake of such foods has less of an effect on health outcomes [128,129,130,131]. The algorithm of the Meiji NPS can also be applied for evaluation per serving size. Evaluating nutritional values both per serving size and per 100 g will ensure a more accurate approach to nutritional intake and health management. Further study is needed to develop and validate the Meiji NPS per serving size.

Spearman’s correlation coefficients for all foods between the Meiji NPS and NRF9.3 were 0.67 for adults and 0.60 for older adults, indicating a moderate correlation [113]. Further, its convergent validity with the HSR also indicated a moderate correlation (r = 0.67 for adults, r = 0.64 for older adults). These results indicate that the Meiji NPSs for adults and older adults are valid NPSs. However, regarding each food group, the correlations between mushrooms, algae, and fish and seafood between the Meiji NPS and NRF9.3 were weak or negligible. This may be explained by the fact that scores in NRF9.3 are calculated per 100 kcal, whereas those in the Meiji NPS are calculated per 100 g. Many foods categorized as mushrooms, fish and seafood, and algae have high water contents, and many others have lower water contents, such as dried mushrooms, dried kelp, and dried horse mackerel. NRF9.3 is not affected by water content, unlike the Meiji NPS, which is demonstrated in these food groups with large variability in water content. In addition, the correlations of eggs between the Meiji NPS and HSR were negligible. In the categories of eggs, processed foods with high sugar content, such as processed sweetened eggs, had high scores in the HSR and low scores in the Meiji NPS. The median scores of pulses, nuts and seeds, vegetables, fruits, mushrooms, algae, fish and seafood, and milk and milk products in the Meiji NPS for adults and older adults were higher than the median scores of total the foods. Diets high in these products have been associated with a lower risk of all-cause and cardiovascular disease-related mortality, as well as dementia, in Japan [132,133,134]. Hence, the Meiji NPS identified foods associated with better health outcomes.

The Meiji NPS may be improved in several ways. Firstly, adding factors such as the quality of nutrients (e.g., amino acid score) and the degree and/or types of food processing (e.g., sterilization, fermentation), which also play important roles in nutritional value [135,136,137], may improve the assessment of foods. NPSs that incorporate such factors have already been proposed [138,139,140]. Secondly, existing NPSs, such as the Nutri-Score and HSR, express scores in an easy-to-understand, 5- or 10-point scale from the perspective of changing consumer behavior [141]. When implementing the Meiji NPS, the score should preferably be expressed in an easy-to-understand manner. For example, the score can be converted into a scale of 0 (least healthful) to 100 (most healthful), improving its interpretability. In fact, we performed this conversion and demonstrated good correlations between the Meiji NPS before and after conversion (Appendix A). Thirdly, additional validation studies can increase the robustness of the Meiji NPS. Specifically, the predictive validity should be determined, that is, the relationship between the Meiji NPS score and the targeted health issues. In addition, affordable and nutrient-rich food products may have an impact on consumers’ health. Nutritional disparities associated with socioeconomic status are also one of the significant issues in Japan [30,31]. Thus, food affordability could be important to consider when using the Meiji NPS. Finally, appropriate revisions may be needed for alignment with changes in Dietary Reference Intakes, official guidelines, and new evidence.

### Limitations

This study has several limitations. Firstly, not all foods were evaluated via the Meiji NPS owing to the lack of nutrient data for a number of foods in the Japanese Food Standard Composition Table 2020 Edition (8th Edition). Secondly, we estimated the volumes of food groups to encourage, as those data are not available in the nutrient composition database. We might have overestimated the nutritional values of the food groups to encourage. Finally, we tested only the convergent validity of the Meiji NPS for adults and older adults.

## 5. Conclusions

The Meiji NPS for adults and older adults was developed based on Japanese health issues that depend on a person’s life stage. The convergent validity between the NRF, HSR, and the Meiji NPS was confirmed. The Meiji NPS may be useful for nutritional evaluation and the reformulation of food products tailored to adults and older adults to ameliorate health issues in Japan.

## Figures and Tables

**Figure 1 nutrients-16-00936-f001:**
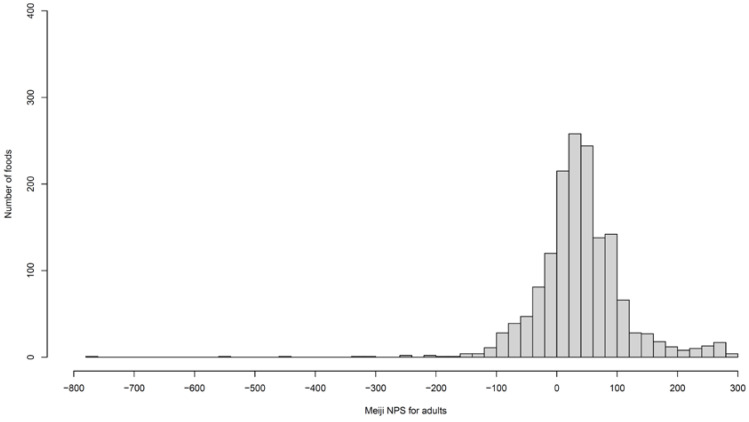
Distribution of the Meiji NPS score for adults. The highest score was 292, and the lowest score was −760. All the outliers (under −200) were seasonings and butters. NPS: Nutritional Profiling System.

**Figure 2 nutrients-16-00936-f002:**
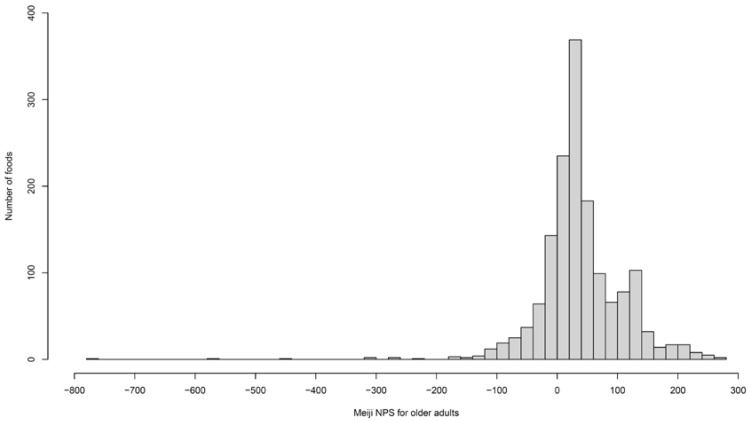
Distribution of the Meiji NPS score for older adults. The highest score was 268, and the lowest score was −770. All the outliers (under −200) were seasoning and butters.

**Table 1 nutrients-16-00936-t001:** Comparison between the Meiji NPS for adults and older adults, NRF9.3, and HSR.

Items	Meiji NPS for Adults	Meiji NPS for Older Adults	NRF9.3	HSR
Nutrients to encourage	Protein Dietary fiber Calcium Iron Vitamin D	Protein Dietary fiber Calcium Vitamin D	Protein Dietary fiber Calcium Iron Potassium Magnesium Vitamin A Vitamin E Vitamin C	Protein Dietary fiber
Food groups to encourage	Fruits Vegetables Nuts Legumes Dairy	Fruits Vegetables Nuts Legumes Dairy	NA ^1^	Fruits Vegetables Nuts Legumes
Nutrients to limit	Energy SFAs Sugar Salt equivalents ^2^	Energy Sugar Salt equivalents ^2^	SFAs Added sugar Sodium	Energy SFAs Total sugar Sodium

^1^ NRF9.3 does not set on any food groups to encourage. ^2^ Sodium content was multiplied by 2.54. NPS: Nutritional Profiling System; NRF: Nutrient-Rich Foods Index; HSR: Health Star Rating; SFA: saturated fatty acid; NA: not applicable.

**Table 2 nutrients-16-00936-t002:** RDVs of the Meiji NPS and NRF9.3.

Items		For Adults	For Older Adults
Nutrients to encourage	Protein	65 g	60 g
	Dietary fiber	21 g	20 g
	Calcium	1000 mg	750 mg
	Iron	12 mg	NA
	Vitamin D	9.5 µg	8.5 µg
Nutrients to limit	Energy	2800 kcal	2400 kcal
	SFAs	31.1 g	NA
	Sugar	70 g	60 g
	Salt equivalents	7.5 g	7.5 g
Food groups to encourage	Fruits	200 g	200 g
	Vegetables	350 g	350 g
	Nuts	75 g	75 g
	Legumes	100 g	100 g
	Dairy	130 g	130 g

RDV: Reference Daily Value; NPS: Nutritional Profiling System; NRF: Nutrient-Rich Foods Index; SFA: saturated fatty acid; NA: not applicable.

**Table 3 nutrients-16-00936-t003:** The caps of the Meiji NPS for adults and older adults.

Items		Meiji NPS for Adults		Meiji NPS for Older Adults	
		Cap	Percentage of RDV	Cap	Percentage of RDV
Nutrients to encourage	Protein	65 g	100%	60 g	100%
	Dietary fiber	21 g	100%	20 g	100%
	Calcium	423.9 mg	42%	389.4 mg	52%
	Iron	5.8 mg	48%	NA	NA
	Vitamin D	6.2 µg	65%	8.5 µg	100%
Nutrients to limit	Energy	NA	NA	NA	NA
	SFAs	NA	NA	NA	NA
	Sugar	NA	NA	NA	NA
	Salt equivalents	NA	NA	NA	NA
Food groups to encourage	Fruits	200 g	100%	113 g	57%
	Vegetables	157.7 g	45%	84.7 g	24%
	Nuts	75 g	100%	75 g	100%
	Legumes	90 g	90%	57 g	57%
	Dairy	108.5 g	83%	55 g	42%

RDVs refer to Dietary the Reference Intakes for Japanese (2020) [6]. The caps were set by the gaps between Dietary Reference Intakes for Japanese (2020) and the National Health and Nutrition Survey [110]. The median intakes of fruits and nuts in adults and the median intake of nuts in older adults were 0 g/day, as reported in the National Health and Nutrition Survey, resulting in the cap of 100%. NPS: Nutritional Profiling System, RDV: Reference Daily Value, NA: not applicable.

**Table 4 nutrients-16-00936-t004:** Summary results of the Meiji NPS for adults.

Items	n	Mean	SD	Median	Max	Min	IQR
Pulses	71	184.9	65.0	169.2	285.6	66.3	127.8 to 253.4
Nuts and seeds	40	163.5	60.6	147.3	292.4	−9.1	129.5 to 196.9
Algae	14	121.1	102.5	152.1	265.6	−73.9	79.0 to 176.0
Mushrooms	46	90.7	68.0	63.1	275.8	−10.9	55.7 to 84.5
Fish and seafood	430	59.7	46.6	63.0	229.3	−155.2	29.6 to 90.8
Vegetables	162	52.5	20.6	46.2	141.7	3.3	39.5 to 65.2
Beverages	10	44.4	102.8	−3.4	251.7	−6.6	−6.4 to −0.7
Milk and milk products	46	40.9	69.4	67.6	186.4	−140.1	15.8 to 83.1
Eggs	15	39.7	37.2	29.8	101.3	−13.2	8.3 to 66.5
Fruits	71	31.4	34.1	40.2	77.7	−165.6	31.2 to 46.4
Potatoes and starches	37	16.7	13.4	15.8	55.3	−12.6	11.1 to 19.7
Cereals	156	11.8	34.6	7.0	133.1	−100.3	−0.2 to 21.4
Meat	303	5.3	40.6	13.7	108.0	−119.3	−15.3 to 33.7
Confectionery	98	−34.6	31.0	−29.6	43.5	−152.2	−52.6 to −12.9
Seasonings and spices	42	−118.7	167.4	−81.3	219.1	−760.3	−127.1 to −43.4
Fats and oils	4	−130.4	73.5	−134.8	−49.2	−202.9	−186.8 to −78.5
Sugars and sweeteners	0	NA	NA	NA	NA	NA	NA
Total	1545	38.9	75.8	36.7	292.4	−760.3	3.2 to 73.1

The Meiji NPS score for adults was calculated for 1545 foods. The food categories followed the food groups in the Japanese Food Standard Composition Table 2020 Edition (8th Edition) [92]. NPS: Nutritional Profiling System; n: number of foods; SD: standard deviation; Max: maximum; Min: minimum; IQR: interquartile range; NA: not applicable.

**Table 5 nutrients-16-00936-t005:** Summary results of the Meiji NPS for older adults.

Items	n	Mean	SD	Median	Max	Min	IQR
Nuts and seeds	40	161.5	44.1	151.9	264.5	106.8	127.8 to 181.3
Pulses	71	132.5	55.8	125.7	220.3	15.2	90.7 to 186.2
Mushrooms	46	86.7	71.7	56.8	268.4	−22.6	50.7 to 81.6
Algae	14	85.9	98.0	110.0	226.7	−115.1	46.5 to 127.8
Fish and seafood	430	69.1	58.4	73.1	226.2	−161.1	26.3 to 119.7
Milk and milk products	46	47.3	44.9	56.3	184.7	−42.4	7.4 to 64.7
Eggs	15	47.0	48.1	28.4	130.5	−19.0	9.4 to 84.2
Vegetables	162	43.5	17.3	39.6	100.8	−6.2	32.1 to 53.8
Beverages	10	35.8	87.5	−6.3	213.1	−7.7	−7.5 to −0.8
Fruits	71	25.2	38.9	36.1	78.8	−173.0	26.6 to 43.2
Meat	303	17.3	19.7	23.3	77.0	−76.9	5.7 to 30.3
Potatoes and starches	37	11.0	14.3	12.0	49.2	−37.6	7.5 to 16.6
Cereals	156	7.5	27.6	3.7	94.5	−106.8	−1.0 to 17.6
Fats and oils	4	−8.1	43.7	−21.8	55.4	−44.4	−29.4 to −0.4
Confectionery	98	−36.3	36.8	−37.3	38.5	−161.3	−60.5 to −9.3
Seasonings and spices	42	−126.1	166.1	−92.1	180.3	−770.2	−131.1 to −45.3
Sugars and sweeteners	0	NA	NA	NA	NA	NA	NA
Total	1545	39.3	70.9	31.2	268.4	−770.2	5.4 to 69.6

The Meiji NPS score for older adults was calculated for 1545 foods. The food categories followed the food groups in the Japanese Food Standard Composition Tables 2020 Edition (8th Edition) [92]. NPS: Nutritional Profiling System; n: number of foods; SD: standard deviation; Max: maximum; Min: minimum; IQR: interquartile range; NA: not applicable.

**Table 6 nutrients-16-00936-t006:** Spearman’s correlation coefficients between the Meiji NPS and NRF9.3.

Items		For Adults		For Older Adults	
	n	r	*p*-Values	r	*p*-Values
Milk and milk products	46	0.91	<0.001	0.81	<0.001
Meat	303	0.91	<0.001	0.72	<0.001
Cereals	156	0.91	<0.001	0.89	<0.001
Beverages	10	0.90	<0.001	0.90	<0.001
Eggs	15	0.84	<0.001	0.79	<0.001
Fats and oils	4	0.80	0.3333	0.40	0.7500
Seasonings and spices	42	0.79	<0.001	0.78	<0.001
Confectionery	98	0.75	<0.001	0.82	<0.001
Fruits	71	0.74	<0.001	0.74	<0.001
Pulses	71	0.68	<0.001	0.68	<0.001
Nuts and seeds	40	0.52	<0.001	0.45	0.0043
Potatoes and starches	37	0.42	0.0110	0.49	0.0019
Vegetables	162	0.39	<0.001	0.41	<0.001
Algae	14	0.24	0.3998	0.20	0.4827
Mushrooms	46	0.14	0.3521	0.07	0.6666
Fish and seafood	430	0.07	0.1239	−0.05	0.2625
Sugars and sweeteners	0	NA	NA	NA	NA
Total	1545	0.67	<0.001	0.60	<0.001

Data for sugars and sweeteners were not available. NPS: Nutritional Profiling System; NRF: Nutrient-Rich Foods Index; NA: not available.

**Table 7 nutrients-16-00936-t007:** Spearman’s correlation coefficients between the Meiji NPS and HSR.

Items	For Adults		For Older Adults	
	r	*p*-Values	r	*p*-Values
Cereals	0.89	<0.001	0.84	<0.001
Meat	0.89	<0.001	0.83	<0.001
Beverages	0.82	0.004	0.79	0.0068
Potatoes and starches	0.75	<0.001	0.64	<0.001
Vegetables	0.74	<0.001	0.72	<0.001
Mushrooms	0.74	<0.001	0.75	<0.001
Seasonings and spices	0.73	<0.001	0.67	<0.001
Algae	0.71	<0.001	0.82	<0.001
Confectionery	0.71	<0.001	0.39	<0.001
Milk and milk products	0.68	<0.001	0.53	<0.001
Pulses	0.60	<0.001	0.75	<0.001
Fruits	0.59	<0.001	0.62	<0.001
Nuts and seeds	0.60	<0.001	0.75	<0.001
Fish and seafood	0.35	<0.001	0.32	<0.001
Eggs	−0.36	0.194	−0.37	0.181
Fats and oils	NA	NA	0.11	0.895
Sugars and sweeteners	NA	NA	NA	NA
Total	0.64	<0.001	0.61	<0.001

NPS: Nutritional Profiling System; HSR: Health Star Rating; NA: not applicable.

## Data Availability

The data presented in this study are openly available in the Japanese Food Standard Composition Table 2020 Edition (8th Edition) at https://www.mext.go.jp/a_menu/syokuhinseibun/mext_01110.html (accessed on 2 February 2024).

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
