# Peer review of "Development and Validation of the Meiji Nutritional Profiling System (Meiji NPS) to Address Dietary Needs of Adults and Older Adults in Japan"

_nutrients, 2024, doi:10.3390/nu16070936_

Round 1
Reviewer 1 Report
Comments and Suggestions for Authors
Thank you for providing the opportunity to review this paper. The focus of this research paper was on developing a new age-sensitive Nutritional Profiling System tailored for Japan and investigating its validity. Given the current research on the ageing population, the findings of this study hold significant importance for public health locally and globally. A few minor issues, however, need to be addressed:
The paper presents crucial insights into the dietary patterns of older adults; however, the title does not accurately reflect the study's objectives. Include “Development and Validation….
I recommend replacing "will be used" with "was used" in line number 84.
Additionally, the recruitment process for the individuals is not clearly stated in the paper, and there is a lack of information on the type of study conducted. However, it appears to be a cross-sectional study.
It is essential to address ethical considerations and consent in the paper. If secondary data were used, it is crucial to clarify whether individuals consented to the original data collection. Also, Information about the type of setting from which the data were collected is unclear and should be specified. Including a section on ethics, consent, and the data collection setting will enhance the transparency and reliability of the study.
There is some redundancy in certain sentences within the method section that could be addressed. Furthermore, it would be beneficial to elaborate on how outliers were handled in the statistical analysis.
Regarding the generalizability of the study, it would be valuable to include comments on this aspect. Moreover, the limitations section could be expanded to cover the lack of generalizability and include suggestions for future directions.
In the discussion section, it is suggested to delve into both groups' demographic and social parameters to provide a rationale for observed differences.
Additionally, social factors such as culture, employment status, and household income, which can impact dietary patterns, are not adequately addressed. It is recommended to include these as limitations.
Hopefully, these suggested revisions and additions will enhance the clarity, completeness, and depth of the paper.
Comments on the Quality of English LanguageI am not the person responsible for assessing the English, but the manuscript appears to be well-written.
Author Response
Thank you for your constructive feedback and suggestions. We appreciate the time and effort you have put into reviewing our paper. Here’s how we plan to address your comments.
The paper presents crucial insights into the dietary patterns of older adults; however, the title does not accurately reflect the study's objectives. Include “Development and Validation….
Reply: We agree that the title includes “validation”. We will make this change in Line 2 in the revised manuscript.
I recommend replacing "will be used" with "was used" in line number 84.
Reply: We agree that “was used” might be more appropriate than “will be used” in this context. We will make this change in Line 84 in the revised manuscript.
It is essential to address ethical considerations and consent in the paper. If secondary data were used, it is crucial to clarify whether individuals consented to the original data collection. Also, Information about the type of setting from which the data were collected is unclear and should be specified. Including a section on ethics, consent, and the data collection setting will enhance the transparency and reliability of the study.
Reply: We would like to clarify that this study does not involve human or animal subjects. Our data analyses were limited to nutrient composition datasets that are publicly available and have been issued by government agencies. We indicate that the data used in this study is sourced from an open database in line 204 in the revised manuscript. Therefore, an ethical review is not required for this study.
There is some redundancy in certain sentences within the method section that could be addressed. Furthermore, it would be beneficial to elaborate on how outliers were handled in the statistical analysis.
Reply: We would like to highlight that outliers are indeed considered an integral part of our results. As mentioned in lines 249 and 253 of our manuscript, we have discussed the role of outliers in our study. Consequently, we have not performed any special treatment or removal of these outliers.
Regarding the generalizability of the study, it would be valuable to include comments on this aspect. Moreover, the limitations section could be expanded to cover the lack of generalizability and include suggestions for future directions.
Reply: In our manuscript, we have indeed focused on the application of the Meiji NPS algorithm for adults and older adults in Japan. However, as we have noted in lines 310 to 313, we think that the algorithm can be applied to other age groups beyond adults and older adults, as well as to populations outside of Japan.
In the discussion section, it is suggested to delve into both groups' demographic and social parameters to provide a rationale for observed differences.
Reply: In this study, we have developed NPSs tailored to adults and older adults to address health issues specific to these life stages. We acknowledge that older adults may have limited resources and access to an affordable and nutritious diet is particularly important for that group. It is difficult to say more at this stage since our research did not involve human subjects or any direct observation of these groups, and as such, we believe the content you proposed is not necessary at this stage. In future research, when we apply the Meiji NPS developed in this study and conduct observational studies on these groups, we will consider incorporating the points you have raised.
Additionally, social factors such as culture, employment status, and household income, which can impact dietary patterns, are not adequately addressed. It is recommended to include these as limitations.
Reply: We would like to clarify that our study does not involve human subjects, and therefore, we are unable to consider social factors. However, we acknowledge that food affordability, measured as the Meiji NPS score per price, could be an important factor for the Meiji NPS in the future. To reflect this, we have added the following sentence to lines 358-361 of our manuscript: “In addition, affordable and nutrient-rich food products may have an impact on consumers’ health. Nutritional disparities associated with socioeconomic status are also one of the significant issues in Japan. Thus, food affordability could be important using the Meiji NPS.”
We hope these revisions will address your concerns and improve the quality of our manuscript. Once again, thank you for your valuable input.
Reviewer 2 Report
Comments and Suggestions for Authors
This is a good quality submission. The Meiji Nutritional Profiling System is explained well and compared to other systems. The split between adults and older adults is original.
In the development of the Meiji NPS, why didn't you include the selection of food groups to DISCOURAGE? Please explain in the manuscript.
line 69 - increase
Author Response
Dear Reviewer2
Thank you for your insightful comments and suggestion. We appreciate your positive feedback on our study. Here’s how we plan to address your comments:
In the development of the Meiji NPS, why didn't you include the selection of food groups to DISCOURAGE? Please explain in the manuscript.
Reply: We would like to clarify that in our study, we have established “food groups to encourage” to account for nutrients that we believe are important and often lacking in the Japanese diet, but could not be included in the “nutrients to discourage”. This point is discussed in lines 117 to 119 of our manuscript. We have added the following sentence to lines 119 to 120 of the revised manuscript: “In alignment with this philosophy, the Meiji NPS did not establish any food groups to discourage.”
line 69 – increase
Reply: We will change the word “increases” to “increase” in Line 69 in the revised manuscript.
We hope these revisions will address your concerns and improve the quality of our manuscript. Once again, thank you for your valuable input.